# Effect of Sewage Sludge Compost Usage on Corn and Faba Bean Growth, Carbon and Nitrogen Forms in Plants and Soil

**Hassan E. Abd Elsalam [1],\*, Mohamed E. El- Sharnouby [1], Abdallah E. Mohamed [2], Bassem M. Raafat [3] and Eman H. El-Gamal [2]**

[1] Department of Biotechnology, College of Science, Taif University, P.O. Box 11099, Taif 21944, Saudi Arabia; m.sharnouby@tu.edu.sa

[2] Soil and Water Technologies Department, Arid Lands Cultivation Research Institute, City of Scientific Research and Technological Applications (SRTA-City), New Borg El-Arab, Alexandria 21934, Egypt; abdallahelsayed44@yahoo.com (A.E.M.); eman.elgamal@yahoo.com (E.H.E.-G.)

[3] Radiological Sciences Department, College of Applied Medical Sciences, Taif University, P.O. Box 11099, Taif 21944, Saudi Arabia; bassemraafat@tu.edu.sa

\* Correspondence: h.abdelsalam@tu.edu.sa

**Abstract:** Sewage sludge is an effective fertilizer in many soil types. When applied as an amendment, sludge introduces, in addition to organic matter, plant nutrients into the soil. When applied for cropland as a fertilizer, the mass loading of sewage sludge is customarily determined by inputs of N and/or P required to support optimal plant growth and a successful harvest. This study aims to examine the changes in organic matter contents and nitrogen forms in sludge-amended soils, as well as the growth of corn and faba bean plants. The main results indicated that there were higher responses to the corn and faba bean yields when sludge was added. Levels of organic carbon in soil were higher after maize harvest and decreased significantly after harvesting of beans, and were higher in sludge amended soils than unmodified soils, indicating the residual effect of sludge in soil. $NO_3^-$-N concentrations were generally higher in the soil after maize harvest than during the plant growth period, but this trend was not apparent in bean soil. The amounts of $NH_4^+$-N were close in the soil during the growth period or after the maize harvest, while they were higher in the soil after the bean harvest than they were during the growth period. Total nitrogen amounts were statistically higher in the soil during the growth period than those collected after the corn harvest, while they were approximately close in the bean soil. The total nitrogen amount in corn and bean leaves increased significantly in plants grown on modified sludge soil. There were no significant differences in the total nitrogen levels of the maize and beans planted on the treated soil.

**Keywords:** sewage sludge; corn; faba bean; nitrogen forms; organic carbon; soil



## 1. Introduction

Sustainable agriculture management practices include recycling nutrients in plants back to future production. Apart from the recycling of beneficial plant nutrients, soil application of biosolids may do away with the requirement for commercial fertilization of cropland [1–3] and can be used as a substitute for chemical fertilizers [4]. Biosolids are an important type of organic waste among the various categories of solid waste [5]. Biosolids, also referred to as sewage sludge (SS) or domestic wastewater residuals, are insoluble biological solid residues or organic waste resulting from different sewage treatment processes [6,7] in wastewater treatment plants worldwide [8].

Sewage sludge is naturally rich in organic carbon, N, P, and micronutrients, which gives it unique fertilizing benefits [9]. Sludge-amended soil differs, to varying extents, from its equivalent unsludged control soil. It tends to have a higher content of organic matter (OM) with variable rates of decomposition, a higher concentration of macro-and micro-nutrients, and the activity of soil microorganisms may be different [7,10]. Sludge is a

good source of N and P for agronomic crops, and many examples of its use on agronomic crops, forest trees, horticulture crops, and plants grown on reclaimed lands have been reported in the literature [11–13].

Land application of SS on agricultural land is now extensively practiced, and one of the most important determinants of application rate is the N supplying capacity of the material applied. The N contained in many sludges is a potentially valuable fertilizer, but due to environmental constraints, it may also be the factor that limits the application rate. As a result, an accurate estimation of the nitrogen mineralization rate is required to ensure the efficient and safe utilization of sludge on agricultural land [14,15]. Usually more than 95% of the N in soil is bonded to C in living and decaying cells of plants, microorganisms, and small animals. This organic-N is not available for plant uptake until it is mineralized to $NH_4^+$ or $NO_3^-$, which supplies a flow of new N that is available for uptake by plants [16].

It is obvious that organic-N mineralization from SS releases $NH_3$, which is the first inorganic N form released. Ammonia is protonated in the soil to form ammonium ion ($NH_4^+$), which has a strong affinity for negatively charged clay and organic particles and is, therefore, less mobile than $NO_3^-$. There is a vast flux of $NH_4^+$ in most soils from mineralization–immobilization turnover, but most of this $NH_4^+$ never leaves the surface horizons of the soils. Ammonium is frequently transported in surface runoff; most commonly in soils that have been treated with organic wastes, such as SS. In most cases, there can be strong physicochemical sinks for $NH_4^+$ on cation exchange sites, and in the interlayer spaces of some clay minerals [17].

Nitrate is the preferred form for the uptake of nitrogen by most plants, and it is usually the most abundant form that can be taken up in well-aerated soils. It usually remains in the soil solution and, therefore, is relatively free to move with water flows [18]. Drainage of excess water often moves $NO_3^-$ downward, in the soil profile, and out of the rhizosphere zone. For these reasons, quantities of $NO_3^-$ in soil usually provide poor estimates of the quantities of N susceptible to uptake during an extended period [17].

Organic matter in soil plays an important role in regulating the behavior of soil pollutants. It provides a high storage capacity of several elements [19]. It is concentrated at the interface between the soil and atmosphere as well as between the soil and plants. Organic carbon (OC) content in soil is a result of a complex biochemical interaction between substrate addition (manure, sludge . . . ) of carbon plant and animal residues and loss of carbon through microbial decomposition and mineralization [20,21].

Soluble carbon enters the soil profile as leachate from living and decaying aboveground phytomass. The concentration of dissolved organic carbon (DOC) retained in a soil-water system depends on the adsorption characteristics of the soil and the organic supply [22,23]. The role of DOC is as the primary resource for biological decomposition in several ecosystems and as the immediately available resource from degrading organic material [22,24]. Loss or retention of soluble organics by soils may be important in controlling nutrient turnover, particularly if the constituents provide a readily available source of energy for soil microbes [24–26].

Sewage sludge is rich in aliphatic and aromatic acids, polysaccharides, proteinaceous material, and organic sulfonates [27,28]. Application of sludge to the soil is expected to affect the content and composition of soil organic carbon (SOC), which, in turn, could influence the speciation, mobility, and fate of trace elements in the amended soil. The incorporation of C-rich sludge into soils has been shown to increase the amount of dissolved organic matter (DOM) in soil [29–31]. Dissolved OM not only mobilizes trace elements, but also affects their bioavailability. Dissolved organic carbon has functional groups that can form chelates with trace elements that can form quite stable complexes [32].

In this study, field experiments were carried out to examine the changes in OM and nitrogen forms in sludge-amended soils cultivated with different plant species. The objectives of this study, therefore, were to investigate the response of two plant species (corn and faba bean) to sludge application and chemical fertilizer (CF) treatments and the

changes in organic carbon and inorganic N forms in sludge amended soils cultivated with corn and faba bean as compared with CF treatments.

## 2. Materials and Methods

### 2.1. Site Description and Studied Soil

Field trials were carried out at Abis lacustrine soil zone, Alexandria, Egypt. The soil of the trials was previously treated with sludge compost and cultivated with different arable crops. The main chemical and physical properties of the soil were determined according to Sparks et al. [33] (Tables 1 and 2). Soil pH was measured in 1:2.5 soil-water suspension after shaking for 30 min, the pH was measured by pH-meter. Electrical conductivity (EC, dS m$^{-1}$) was measured in 1:1 soil-water extract using a conductivity meter. Total organic carbon (TOC, g kg$^{-1}$) using wet oxidation with dichromate in sulphuric acid; total-N (g kg$^{-1}$) using the Kjeldahl method. The total content of macro-and micronutrients, and heavy metals in the soil and SS was determined after combustion of OM, digesting the samples in a mixture (3:2, *v/v*) of 70% $HClO_4$ and 65% $HNO_3$, and analyzing the filtrates using atomic emission spectrometer [34]. Two plant species were used for field studies, faba bean (*Vicia faba* L.) variety *Renablanka* as a winter plant and corn (*Zea maize* L.) variety 10, a single hybrid as a summer plant.

### 2.2. Sludge Compost

Sludge compost was obtained from Site 9N, Alexandria General Organization for Sanitary Drainage (AGOSD), Alexandria, Egypt. The chemical and physical characteristics of the used SS compost were measured as described before in studied soil analysis [33,34], and data are given in Tables 3 and 4.

### 2.3. Chemical Fertilizers

The chemical fertilizers (CF) used in this study were (i) ammonium nitrate (33.5% N), (ii) single superphosphate (15.5% $P_2O_5$), and (iii) potassium sulfate (52% $K_2O$).

**Table 1.** The main chemical and physical characteristics of the soil.

| Soil Parameters | |
|---|---|
| Reaction (pH-unit *) | 7.96 |
| Electrical Conductivity (EC **, dS m$^{-1}$) | 0.67 |
| Cation Exchange Capacity (CEC, Cmol$_{(+)}$ kg$^{-1}$) | 25.9 |
| **Water-soluble ions **** | |
| $Ca^{++}$ (mg kg$^{-1}$) | 31.6 |
| $Mg^{++}$ (mg kg$^{-1}$) | 32.0 |
| $Na^+$ (mg kg$^{-1}$) | 43.5 |
| $K^+$ (mg kg$^{-1}$) | 18.7 |
| $HCO_3^-$ (mg kg$^{-1}$) | 137.3 |
| $Cl^-$ (mg kg$^{-1}$) | 62.1 |
| $SO_4^{--}$ (mg kg$^{-1}$) | 124.8 |
| Sodium Adsorption Ratio (SAR) | 1.30 |
| Organic matter (OM, %) | 0.84 |
| Organic carbon (OC, %) | 0.49 |
| Total nitrogen (TN, %) | 0.06 |
| C/N ratio | 8.17 |
| Total carbonates ($CaCO_3$, %) | 13.6 |
| Water holding capacity (%) | 56.2 |
| Bulk density (kg dm$^{-3}$) | 1.29 |
| Texture | Sandy loam |

* In 1:2.5 Soil: Water ratio, ** In 1:1 soil: water extract.

**Table 2.** Elemental composition of the studied soil.

| Macronutrients (mg kg$^{-1}$) | Total | Bioavailable |
|---|---|---|
| Nitrogen (N) | 600 | — |
| $NH_4^+$-N | — | 9.84 |
| $NO_3$-N | — | 8.78 |
| Organic-N | 581 | — |
| Phosphorous (P) | 3200 | 10.5 |
| Potassium (K) | 2000 | 461 |
| **Micronutrients (mg kg$^{-1}$)** | **Total** | **Bioavailable** |
| Cu | 15.2 | 0.67 |
| Fe | 887 | 16.9 |
| Mn | 642 | 5.2 |
| Zn | 56 | 9 |
| **Heavy metals (mg kg$^{-1}$)** | **Total** | **Bioavailable** |
| Cd | 7.4 | 0.05 |
| Cr | 19.0 | 0.22 |
| Ni | 29.8 | 0.64 |
| Pb | 20.0 | 2.87 |

(—) not measured.

**Table 3.** The main chemical and physical characteristics of the used composted sewage sludge. Data are reported on a dry matter basis.

| Parameters | |
|---|---|
| Reaction (pH-unit) | 6.39 |
| Electrical Conductivity (EC, dS m$^{-1}$) | 23.4 |
| **Water-soluble ions (In 1:1 water: sludge)** | |
| $Ca^{++}$ (mg kg$^{-1}$) | 1860 |
| $Mg^{++}$ (mg kg$^{-1}$) | 516 |
| $Na^+$ (mg kg$^{-1}$) | 1633 |
| $K^+$ (mg kg$^{-1}$) | 1026 |
| $HCO_3^-$ (mg kg$^{-1}$) | 3538 |
| $Cl^-$ (mg kg$^{-1}$) | 5066 |
| $SO_4^{--}$ (mg kg$^{-1}$) | 163 |
| Sodium Adsorption Ratio (SAR) | 8.61 |
| Organic matter (OM, %) | 35.3 |
| Organic carbon (OC, %) | 20.5 |
| Total nitrogen (TN, %) | 1.76 |
| C/N ratio (%) | 11.62 |
| Total carbonates (CaCO3, %) | 18.0 |
| Water holding capacity (kg dm$^{-3}$) | 76.8 |
| Bulk density (%) | 0.75 |

*2.4. Experimental Trials*

A field trial was carried out in the summer growing season (June 2018) to investigate the residual effect of different rates of added sludge as well as the cumulative amounts formerly applied on corn plants and soil. The winter growing season (2018/2019) was carried out to investigate the cumulative residual effect of sludge on the properties of soil and growth of faba beans. The field trials were planned as a factorial experiment with two factors in a randomized complete block design (RCBD), with six treatments of sludge compost and NPK-CF and in four replicates.

Treatments were carried out as follows: (i) The sludge (SS compost) applications history to the soils is shown in Table 5. (ii) The NPK-CF were applied as follows: 200 kg/acre superphosphate fertilizer was added to the soil of the two crops before planting, 100 kg/acre ammonium nitrate fertilizer was added to faba bean soil during the growing season, while

300 kg/acre fertilizer was applied to the corn soil during the growing season in two equal doses, and 100 kg/acre potassium sulfate fertilizer was added to the soil during the growing season in one dose.

**Table 4.** Elemental composition of the used composted sewage sludge.

| Macronutrients (mg kg$^{-1}$) | Total | Bioavailable |
|---|---|---|
| Nitrogen (N) | 17,590 | — |
| NH$_4^+$-N | — | 854 |
| NO$_3$-N | — | 295 |
| Organic-N | 16,441 | — |
| Phosphorous (P) | 1743 | 220 |
| Potassium (K) | 4765 | 0.93 |
| **Micronutrients (mg kg$^{-1}$)** | **Total** | **Bioavailable** |
| Cu | 133.8 | 40.0 |
| Fe | 7296.0 | 384 |
| Mn | 169.6 | 106 |
| Zn | 3536.0 | 220 |
| **Heavy metals (mg kg$^{-1}$)** | **Total** | **Bioavailable** |
| Cd | 12.4 | 0.38 |
| Cr | 37.2 | 0.94 |
| Ni | 21.2 | 2.70 |
| Pb | 144.8 | 32.8 |

**Table 5.** The sludge application (t/acre) history and the treatments carried out in the field trials during the period 2017/2019.

| Treatment No. | Summer 1st Season | Winter 1st Season | Summer 2nd Season | Winter 2nd Season | Cumulative Sludge Applied | NPK-CF |
|---|---|---|---|---|---|---|
| 1 | 0 | 0 | 0 | 0 | 0 | — |
| 2 | 0 | 0 | 0 | 0 | 0 | Full dose |
| 3 | 25 | 50 | 50 | 0 | 125 | — |
| 4 | 25 | 50 | 50 | 0 | 125 | Half dose |
| 5 | 50 | 100 | 100 | 0 | 250 | — |
| 6 | 50 | 100 | 100 | 0 | 250 | Half dose |

Planting was carried out as follows: (i) The corn was planted on 6 June 2018. The treatments carried out were as follows: (a) 0, 80, and 160-t sludge/acre. (b) 0, 80, and 160-t sludge/acre plus half-dose NPK-CF. (ii) The faba bean was planted on 20th November 2019. In this season, no sludge was added.

The residual effect of sludge in soils from the previous application (Table 5) was considered as the sludge treatments. The objective, therefore, was to study the residual effect of previous repeated sludge applications. In the summer field trial (2018), corn was planted in the soil after faba beans (winter crop, 2017/2018), while in the winter field trial, faba beans were cultivated after corn (summer crop 2018).

*2.5. Soil Samples*

Soils planted with corn and faba beans were collected at two depths (0–15 and 15–30 cm) as follows: Before planting, after 55 days from faba bean planting, after 65 days from corn planting, and after the two crops were harvested. The soil samples were air-dried, ground, and passed through a 2 mm sieve, and stored for chemical and physical analysis.

### 2.6. Plant Samples of Corn and Faba Bean

Plant samples were collected according to the methods outlined by Chapman and Pratt [35], as the highest three leaves of corn plant were collected after 65 days from seeding, collected the whole above-ground part of faba bean plant (shoot) after 55 days from planting, and grains and stover of both corn and faba bean were collected at harvest. The dry weight of the different plant parts of corn and faba bean were measured.

### 2.7. Soil Analysis
2.7.1. Organic C and DOC Analysis in Soils

Organic C was determined in the soil by the spectroscopic method [36], and DOC was determined after extraction [37].

2.7.2. Extraction and Determination of Soils N forms

Total and mineral nitrogen forms were extracted [extraction at 1:5 with 2M potassium chloride (KCl)] according to the method described by Forster [38] and determined all nitrogen forms, Total-N, $NH_4^+$-N, $NO_2^-$-N according to the method described by Sparks et al. [33], and $NO_3^-$-N by modified Griess Ilosvay method [39].

### 2.8. Statistical Analysis

The experiment was arranged in a randomized complete block design (RCBD) with four replicates. Data were analyzed by two-way analysis of variance (ANOVA) at $p \leq 0.05$ using statistical functions of Co-Stat software for statistics (2004). Further, a least significant difference (LSD $_{0.05}$) test was used to differentiate between significant and non-significant means.

## 3. Results and Discussion
### 3.1. Characteristics of the Soil and Sludge

The experimental soil of ARS is a lacustrine deposit. As shown in Table 1, the soil is non-saline with an average EC of 0.67 dS m$^{-1}$ and is non-alkaline with a SAR value of 1.30. The soil has a pH value of 7.96. Additionally, the soil has low OM content (an average of 0.84%) and low total-N content (an average of 0.06%) and C/N of 8.17, and a relatively high level of total carbonate (an average of 13.6%). The soil used has a sandy loam texture, bulk density with an average of 1.29 kg dm$^{-3}$, and water holding capacity (WHC) average of 56.2% (Table 1).

As shown in Table 2, the amounts of available N and organic-N were about 3.1% and 96.9% of total N, respectively. Nitrogen bioavailable from SS is reported to range between 0% and 56% [40]. The bioavailable P (10.5 mg kg$^{-1}$) is at a medium level (8–14 mg kg$^{-1}$), and the level of available K is high (>450 mg kg$^{-1}$) [41]. The concentrations of total trace elements (Cu, Fe, Mn, Zn, Cd, Cr, Ni, Pb) in soils are within the normal range reported by Elsokkary and Låg [42] and Kabata-Pendias [43].

According to DTPA (standard method), and AB-DTPA extraction method [44,45], the data in Table 2 show that there are normal levels of bioavailable (AB-DTPA) Cu, Fe, Mn, and Zn in the soil as compared with Standard Method and AB-DTPA extraction. This indicates that the soil did not suffer micronutrient deficiency.

The sludge readily achieved thermophilic composting temperature between 55 and 65 °C. The temperature of compost is stable within that range for up to 15 days. The sludge is composted aerobically. This process will generate enough heat only if it can be kept sufficiently aerobic employing some form of aeration. Thus, "Windrow or Aerated piles" composting is used with the frequent turning of the pile of compost which aerates it. The process of sludge composting was carried out at site 9N of AGOSD where the used sludge compost was obtained.

The data in Table 3 shows that the sludge compost used in this study had a slightly acidic reaction (pH 6.39), but is highly saline (EC 23.4 dS m$^{-1}$). This sludge contains relatively low OM, which is due to dumping the sludge on the calcareous soil that was

used for drying the raw watered sludge. The bulk density of SS was 0.75%, which develops a good soil structure when added to soil. Ramesh et al. [46] reported that the application of organic fertilizers reduced bulk density of soil in organic farms, which indicates better soil aggregation and soil physical conditions. Soil physical properties can be improved by the incorporation of optimum OM level waste. For example, in the saline-sodic soil of cold regions, soil physical properties were improved by the application of organic wastes, which reduced bulk density, increased aggregate stability and permeability, and thus restore this type of soil [47]. Similarly, soil organic carbon (SOC) also stimulates hydraulic conductivity by improving aggregate stability and porosity of agricultural soils [48], which can be managed by applying organic wastes [49]. The addition of manure showed similar trends in reducing bulk density, increasing porosity and SOC content, and thus soil physical properties improvement [50–52]. Additionally, Yazdanpanah et al. [53] proved that the incorporation of solid wastes increases SOC content in soil and significantly increased water stable aggregates and total porosity, and thus influenced the soil texture.

As a result, the sludge contained a relatively high carbonate percentage (18.04%) due to dumping the sludge on the calcareous soil that was used for drying the raw watered sludge. It contains a relatively low amount of total-N (1.76%), which is lower than the normal range (2.6%) reported in the literature, and high content of organic carbon (20.45%). These results were in line with previous findings that municipal SS constituted a rich source of OM and other biogenic elements [54,55].

As shown in Table 3, C/N ratio was less than 20% (11.62%), a satisfactory ratio, and will not give rise to N immobilization processes when land applied. This is because C/N is a major factor affecting microbial activity in composting biomass, which in turn affects OM degradation and nitrogen losses.

Most of the inorganic N in the sludge was in the $NH_4^+$ form (4.86%) while $NO_3^-$ consisted of only about 1.68% of the total-N. The level of $NO_2^-$-N was not detected (Table 4). The total P content of the sludge was about 0.17% which is considered low as compared with the usual level (0.8–6.1%) [56]. However, a relatively high proportion of the total P of the sludge was extracted with AB-DTPA and represented 2.62% of the total P (Table 4).

Analyses SS compost showed that it contains satisfactory quantities of micronutrients Cu, Fe, Mn, Zn, and Ni (Table 4). Micronutrients, for many years, have been considered as less essential in plant nutrition. Presently, it is well documented that micronutrients play an important role in increasing crop yield through their involvement in the metabolism of macronutrients, as well as crop responses to environmental stress conditions [57]. Historically, micronutrients were only used with natural organic fertilizers, such as manure, slurry, green manure, or compost. Currently, the most common sources of micronutrients are synthetic and mineral fertilizers. A wide range of these fertilizers are available commercially, so it is very easy to find a suitable product to meet the requirements of plants. Unfortunately, these fertilizers are very expensive. Moreover, at the present time, regarding the principles of sustainable agriculture and environmental protection, farmers are paying more attention to the possibility of using organic fertilizers, which are in line with consumer expectations. The cited authors proved that systematic long-term application of organic fertilizers not only results in an increased availability of soil OM, but also in an increase of micronutrient amounts available to plants.

The content of trace elements in SS is one of the limiting factors for using SS as a soil amendment. The limits for Cd, Cr, Cu, Ni, Pb, and Zn have been established by European legislation (EU Directive 86/278/EEC) and represent the maximum permitted concentrations in SS for agricultural use [58]. Table 4 shows that the concentrations of trace metals in the SS samples are below EU threshold limits. Accordingly, the studied SS could be safely applied to soil according to current legislative-based recommendations. However, caution should be advised and an assessment of the long-term impacts of soil amendment with MSS is recommended.

### 3.2. Plant Growth

The responses of the different plant species to NPK-CF with or without biosolid application to soils are outlined as follows. As shown in Table 6, the growth yield of plants, whether at a certain growth stage or the harvest stage, i.e., stover and grain yield, was generally higher than the yield of the control that did not receive any treatment. Thus, this data represents the response of a crop to treatments applied to the soil.

**Table 6.** Effect of soil treatments on dry weight of corn leaves (65 days after seeding) and biomass yield after harvest.

| Treatment | | Leaves Dry Weight (g/One Leaf) | Stover (t/Acre) | Grain (t/Acre) | Biomass (t/Acre) | Grain % of Biomass | Stover/Grain Ratio |
|---|---|---|---|---|---|---|---|
| Sludge (t/Acre) | NPK-Chem. Fert. | | | | | | |
| 0 | 0 | 1.32 | 7.56 | 1.66 | 9.33 | 17.8 | 4.55 |
| 0 | NPK | 1.82 | 11.13 | 1.95 | 13.07 | 14.9 | 5.71 |
| 80 | 0 | 1.10 | 11.68 | 3.15 | 14.83 | 21.2 | 3.71 |
| 80 | 1/2 NPK | 1.35 | 12.28 | 4.39 | 16.67 | 26.3 | 2.80 |
| 160 | 0 | 0.95 | 12.01 | 5.13 | 17.13 | 29.9 | 2.34 |
| 160 | 1/2 NPK | 1.34 | 11.56 | 5.91 | 17.47 | 33.8 | 1.96 |
| LSD$_{0.05}$ | | 0.32 | 0.34 | 0.26 | 0.39 | — | — |

#### 3.2.1. Corn Plant Growth

As shown in Table 6, there are significant increases in leaf dry weight of corn plant as a result of NPK-CF application relative to the control plant (0 NPK-CF and 0 sludge). However, sludge application did not significantly affect dry weight of corn leaf. Additionally, there were significant increases in the stover, grain, and biomass yield with NPK-CF and with sludge application. The maximum grain yield was obtained when soil was amended with 160 t/acre of sludge incorporation with half a dose of recommended NPK-CF (5.91 t/acre), and it contributed to raise grain yield by 200% over the full NPK-CF fertilized plants. In respect of values of the stover/grain, they had decreased with a higher sludge rate. This indicates that sludge application increased the grain yield relative to that of the stover. Similar results were obtained in Saudi Arabia [59], in Spain [60], and in Tunisia [61]. In consequence, these authors observed that sludge addition positively impacted soil fertility. This in turn may positively enhance crop yield when grown on sludge-amended soils.

#### 3.2.2. Faba Bean Plant Growth

As shown in Table 7, there were no significant variations in leaf dry weight, while there were significant increases in the stem dry weight of 55-days old faba bean plants, especially at a higher sludge rate. Additionally, there were significant increases in stover, grain, and biomass yield with sludge application. The application of sludge alone (160 t/acre) significantly increased grain yield (2.41 t/acre) more than grain yield after application of full dose of NPK-CF (1.69 t/acre), and there was no significant increase in grain yield when NPK-CF were added with sludge (2.45 t/acre). Therefore, we can use sludge and reduce CF application with significant promotion of faba bean growth more than application of 100% NPK-CF to save chemical fertilizer application and maintain environment and soil health.

In general, plants amended with sewage sludge compost performed significantly ($p < 0.05$) better than the plants having chemical fertilizer alone. This may be attributed to the ability of sludge to improve soil properties and availability of essential elements for plant growth promotion. This suggests that incorporation of sewage sludge compost with the addition of small amounts of inorganic chemical fertilizers can be used as a feasible alternative to increase the crop production in an environmentally friendly way. Long term experiments comparing productivity and soil health parameters at ICRISAT have demonstrated that organic practices produced yields comparable to conventional plots, without receiving any chemical fertilizer; they actually showed an increase in the



concentration of macro-elements compared with the conventional. In another similar study, results showed improvements of different magnitudes in respect to soil organic carbon, available-P, available-K, bulk density, and microbial activity under organic systems as compared to chemical farms [62].

**Table 7.** Effect of soil treatments on dry weight of faba bean leaves and stems (55 days after seeding) and biomass yield after harvest.

| Treatment | | Dry Weigh (g/One Plant) | | Stover (t/Acre) | Grain (t/Acre) | Biomass (t/Acre) | Grain % of Biomass | Stover/Grain Ratio |
|---|---|---|---|---|---|---|---|---|
| Sludge (t/Acre) | NPK-Chem. Fert. | Leaves | Stems | | | | | |
| 0 | 0 | 1.31 | 1.18 | 4.26 | 1.68 | 5.93 | 28.3 | 2.54 |
| 0 | NPK | 1.81 | 1.34 | 4.31 | 1.69 | 6.01 | 28.1 | 2.55 |
| 80 | 0 | 1.94 | 1.49 | 5.07 | 1.99 | 7.06 | 28.2 | 2.55 |
| 80 | 1/2 NPK | 1.79 | 1.37 | 5.04 | 2.15 | 7.19 | 29.9 | 2.34 |
| 160 | 0 | 2.42 | 2.97 | 5.46 | 2.41 | 7.87 | 30.6 | 2.27 |
| 160 | 1/2 NPK | 2.06 | 1.75 | 5.52 | 2.45 | 7.97 | 30.7 | 2.25 |
| LSD$_{0.05}$ | | N.S.* | 0.40 | 0.12 | 0.08 | 0.14 | — | — |

\* (N.S) Not significant.

### 3.3. Organic Carbon in Corn Soils

The effect of biosolid and NPK-CF treatments on OC contents in soils collected after 65 days from corn planting and after crop harvest (Table 8) showed that the levels of SOC significantly increased with sludge application up to 160 t/acre, relative to control soil (1.61% and 0.74%, respectively). While there was no significant effect of NPK-CF treatment on the level of OC in soil (0.8%) relative to the control soil.

**Table 8.** Effect of soil treatments on organic carbon (%) in corn soil after 65 days from seeding and after crop harvest.

| Soil Depth (cm) | Treatment | | | | | | LSD$_{0.05}$ |
|---|---|---|---|---|---|---|---|
| | Sludge | 0 | 0 | 80 | 80 | 160 | 160 | |
| | CF | 0 | NPK | 0 | 1/2NPK | 0 | 1/2NPK | |
| 65 days after seeding | | | | | | | |
| 0–15 | | 0.74 | 0.80 | 1.37 | 1.40 | 1.42 | 1.61 | 0.27 |
| 15–30 | | 0.70 | 0.73 | 0.89 | 1.31 | 1.31 | 1.29 | |
| LSD$_{0.05}$ | | 0.19 | | | | | | |
| After crop harvest | | | | | | | |
| 0–15 | | 0.88 | 0.88 | 1.66 | 1.80 | 2.94 | 3.81 | 0.63 |
| 15–30 | | 0.84 | 0.83 | 1.05 | 1.34 | 1.95 | 2.05 | |
| LSD$_{0.05}$ | | 0.45 | | | | | | |

Incorporation of NPK-CF with sludge amended soil had significantly increased SOC. This could be due to the decomposition of sludge OM, which might be stimulated with NPK-CF [63,64]. Table 8 showed that SOC levels were higher in soil samples collected after corn harvest relative to those collected during the growth season. This increase was attributed to the effect of corn plant residue in the soil, which affects the decomposition of sludge in soils [65]. Generally, the levels of SOC significantly decreased with soil depth.

### 3.4. Organic Carbon in Faba Bean Soils

The residual effects of biosolid and NPK-CF treatments on the levels of OC in soils collected 55 days after faba bean seeding and after crop harvest were shown in Table 9. There are significant increases in the levels of SOC in sludge-amended soils relative to those of the NPK-CF and control soils. The levels of OC were lower in the sludge-amended soils

in samples collected after faba bean harvest than those collected after 55 days from planting. These levels were also markedly lower than those of the NPK-CF soils. Additionally, the levels of OC were decreased with soil depth.

**Table 9.** Effect of soil treatments on organic carbon (%) in faba bean soil after 55 days from seeding and after crop harvest.

| Soil Depth (cm) | Treatment | | | | | | | LSD$_{0.05}$ |
|---|---|---|---|---|---|---|---|---|
| | **Sludge** | **0** | **0** | **80** | **80** | **160** | **160** | |
| | **Ch. Fe.** | **0** | **NPK** | **0** | **1/2NPK** | **0** | **1/2NPK** | |
| | *55 days after seeding* | | | | | | | |
| 0—15 | | 1.21 | 1.13 | 1.89 | 1.79 | 2.20 | 2.06 | 0.34 |
| 15—30 | | 0.96 | 0.95 | 1.16 | 1.17 | 1.78 | 1.66 | |
| LSD$_{0.05}$ | | 0.24 | | | | | | |
| | *After crop harvest* | | | | | | | |
| 0—15 | | 1.32 | 1.02 | 1.50 | 1.46 | 2.12 | 2.14 | 0.28 |
| 15—30 | | 0.91 | 1.01 | 1.16 | 1.24 | 1.74 | 1.45 | |
| LSD$_{0.05}$ | | 0.20 | | | | | | |

It has been reported by several researchers that repeated application of SS compost in agricultural land helps in increasing the OM content and C/N ratio of soil in comparison to unamended soil [66–68]. This helps in maintaining soil fertility and its productivity. Therefore, organic fertilizer (like SS compost) could be considered as a promising and sustainable alternative to inorganic fertilizer in agriculture.

Effectively managing organic waste in our soils is the best way to boost SOC through an economical approach. Besides, this also enhances the carbon sequestration potential [69]. One of the best organic waste management techniques is composting, the process of converting organic wastes into more stable C, including humus-like substances, which is rich in nutrients. Microorganisms in the presence of air generally degrade complex waste materials and produce compost, carbon dioxide, and water [70]. Alternately, SS application on land is the best economical practice, which in turn increases the water holding capacity, aeration, and OM and soil nutrients [71–73].

### 3.5. Nitrogen Forms in Corn Soils

Concentrations of $NO_3^-$, $NO_2^-$, $NH_4^+$, and total-N in soil samples collected after 65 days from corn planting and after corn harvest are shown in Figure 1. After 65 days, the levels of $NO_3^-$ ion concentration significantly increased in the sludge amended soils relative to the control and the NPK-CF soils. The presence of NPK-CF, with or without sludge, significantly decreased $NO_3^-$ concentration in the soil relative to the sludge-amended soil alone. It reflects the eluted available N in the form of $NO_3^-$-N derived from N mineralization of added sludge. This indicates that $NO_3^-$ ions may be lost, in the presence of NPK-CF, through ammonification or by transformation to $N_2O$ or by evaporation as $NO_2$.

The $NO_3^-$ concentration was lower in the lower soil layer (15–30 cm) than those in the upper soil layer (0–15 cm). This shows that there was no leaching of $NO_3^-$ downward soil depth and indicated that the oxygen concentration was optimal for nitrification in the upper soil layer than the lower soil layer because this is an aerobic process. It is also clear that the levels of $NO_3^-$ of the low sludge soil (80 t/acre) were significantly lower than those of the high sludge soil (160 t/acre) of both the upper or lower soil layers. There were no significant differences in the amount of $NO_2^-$ in soils of the different treatments. Similarly, there were no significant differences in the levels of $NH_4^+$ in soils receiving different treatments. However, there were significantly higher levels of $NH_4^+$ in the lower soil layer (15–30 cm) than in the upper soil layer (0–15 cm). Additionally, there were significant increases in total-N in soils treated with sludge relative to the control and NPK-CF soils.

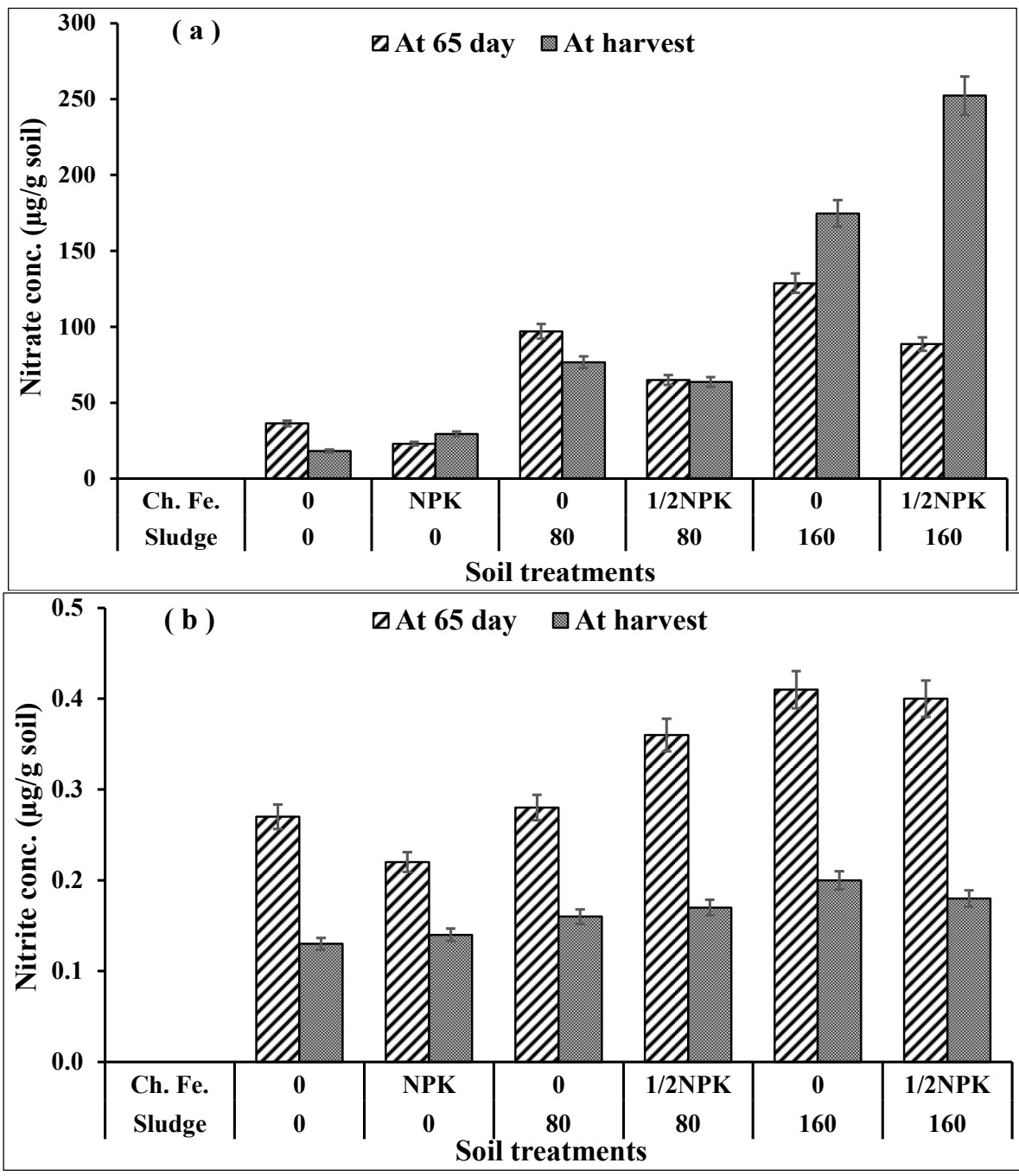

**Figure 1.** *Cont.*

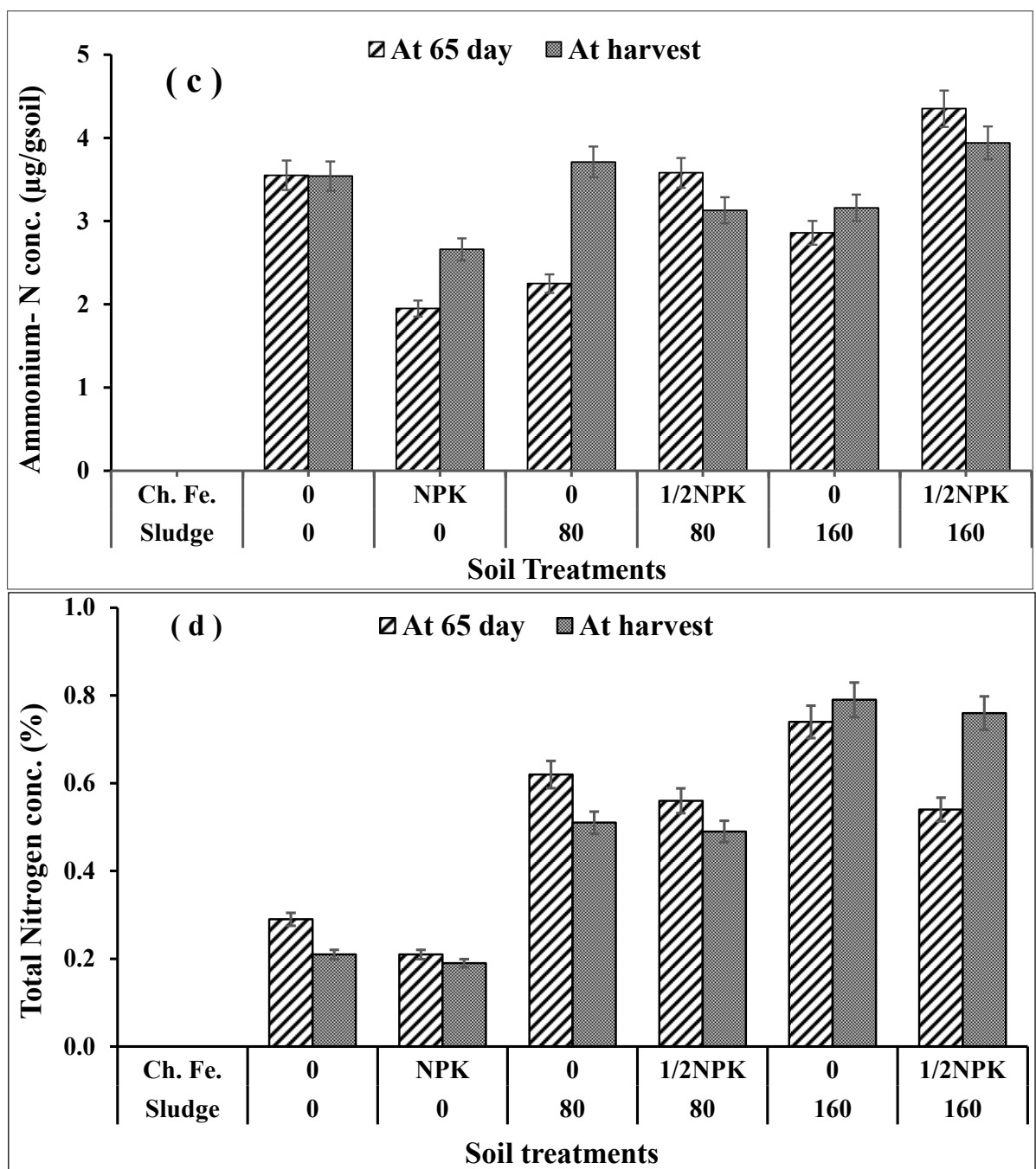

**Figure 1.** Effect of soil treatments on $NO_3^-$-N (**a**), $NO_2^-$-N (**b**), $NH_4^+$-N (**c**), and total nitrogen (**d**) in corn soil after 65 days from seeding and after harvest.

In respect, after corn harvest, there are significant increases in the levels of $NO_3^-$ in sludge-amended soil. A higher sludge rate produced higher $NO_3^-$ concentration in soils. The concentrations of $NO_3^-$ were significantly higher in the upper soil layer (0–15 cm) than the lower layer (15–30 cm). This indicates that there was no leaching of $NO_3^-$ downward with soil depth. The concentrations of $NO_2^-$ in soils significantly varied among the soils of the different treatments. Additionally, there were no significant differences in the levels of $NH_4^+$ in soils of the different treatments, which could be due to volatilization of ammonium in the studied soil with high pH and $CaCO_3$ content (Table 1), and conversion of ammonium to nitrate through the nitrification process [74]. Li and Li [75] found that the concentration of ammonium in animal-manure-treated soil was reduced within two

weeks due to a rapid nitrification process. They also reported that microorganisms only use nitrates when the concentration of ammonium is less than 1 mg N kg$^{-1}$ soil. In fact, microorganisms prefer nitrogen in ammonium form for their growth and proliferation. Additionally, Yousif and Mubarak [56] found a reduction of $NH_4^+$-N during the first two weeks of incubation due to $NH_3^+$ volatilization and rapid nitrification. However, there were significant differences in the levels of $NH_4^+$ between the soils of the upper and lower layers. There were also significantly higher levels of $NH_4^+$ in lower soil layer (15–30 cm) than those of the upper soil layer (0–15 cm).

The concentrations of total-N were significantly higher in the sludge-amended soils than those of the control, and NPK-CF soils and were higher in soils treated with higher sludge rates, and these results are in agreement with Vaca et al. [76]. The levels of total-N were significantly higher in the soils of the upper layer (0–15 cm) than those of the lower layer (15–30 cm). Most studies linked enhancement of soil OM and nitrogen status to amendment rates [77–79].

In the present study, $NO_3^-$-N concentration was the dominant form of inorganic-N, a result that agreed with the findings reported by Yousif and Mubarak [56]. They concluded that this increase could be due to alkaline nature of the soil pH (>7.0) similar to our soil pH (Table 1) and may be more favorable for nitrifying bacteria than the ammonifying bacteria [80]. Additionally, Alizadeh et al. [81] found the same results on $NO_3^-$-N concentration in soils amended with SS.

Organic manures included several types of N compounds which have various resistances to analysis by microorganisms that resulted in the trend of $NO_3^-$-N releasing during the incubation time periods [82]. Li and Li [75] studied the N mineralization of cattle, chicken, and pig manures and found that the $NO_3^-$-N concentration rapidly increased during the 56–84 days after application and afterward increased slowly. In another study, $NO_3^-$-N concentration in the SS-amended soil increased sharply after 28 days and did not change significantly thereafter [83]. This trend of release of $NO_3^-$-N with all manure- and SS-treated soil samples should be considered for manure management under field conditions and for crop production at farm levels to increase N uptake efficiency and minimizing $NO_3^-$-N loss. Additionally, Randall and Sawyer [84] suggested that to obtain high N-use efficiency (NUE) of applied manure and minimize potential N losses via nitrification and denitrification, the nitrogen availability and rate of mineralization during the growing season should be considered.

### 3.6. Nitrogen Forms in Faba Bean Soils

The concentration of $NO_3^-$, $NO_2^-$, $NH_4^+$, and total-N in soil samples collected after 55 days from faba bean planting and after harvest are illustrated in Figure 2.

After 55 days from seeding, the levels of $NO_3^-$ significantly increased in the sludge-amended soil at a residual sludge rate of 160 t/acre, relative to the other treatments. Higher significant levels of $NO_3^-$ were found in the upper soil layer than the lower soil layer. $NO_2^-$ concentrations significantly decreased in sludge-amended soil at a sludge rate of 80 t/acre. While there is a significant increase only in sludge-amended soils incorporated with NPK-CF in soils of the lower layer (15–30 cm) than those of the upper layer (0–15 cm). The level of $NH_4^+$ significantly decreased in the sludge-amended soils at 80 t/acre sludge rate while it significantly increased at the higher sludge rate (160 t/acre) relative to the control and NPK-CF soils. Incorporation of NPK-CF with sludge significantly decreased $NH_4^+$ level in soil at 160 t/acre sludge rate. Generally, the concentrations of $NH_4^+$ in soils significantly decreased with the depth of the soil. The soil contents of total-N significantly increased in the sludge-amended soils relative to the control soil. It is clear that NPK-CF incorporated with sludge significantly decreased the total-N of soils. Generally, there were no significant differences in the levels of total-N in the soil layers.

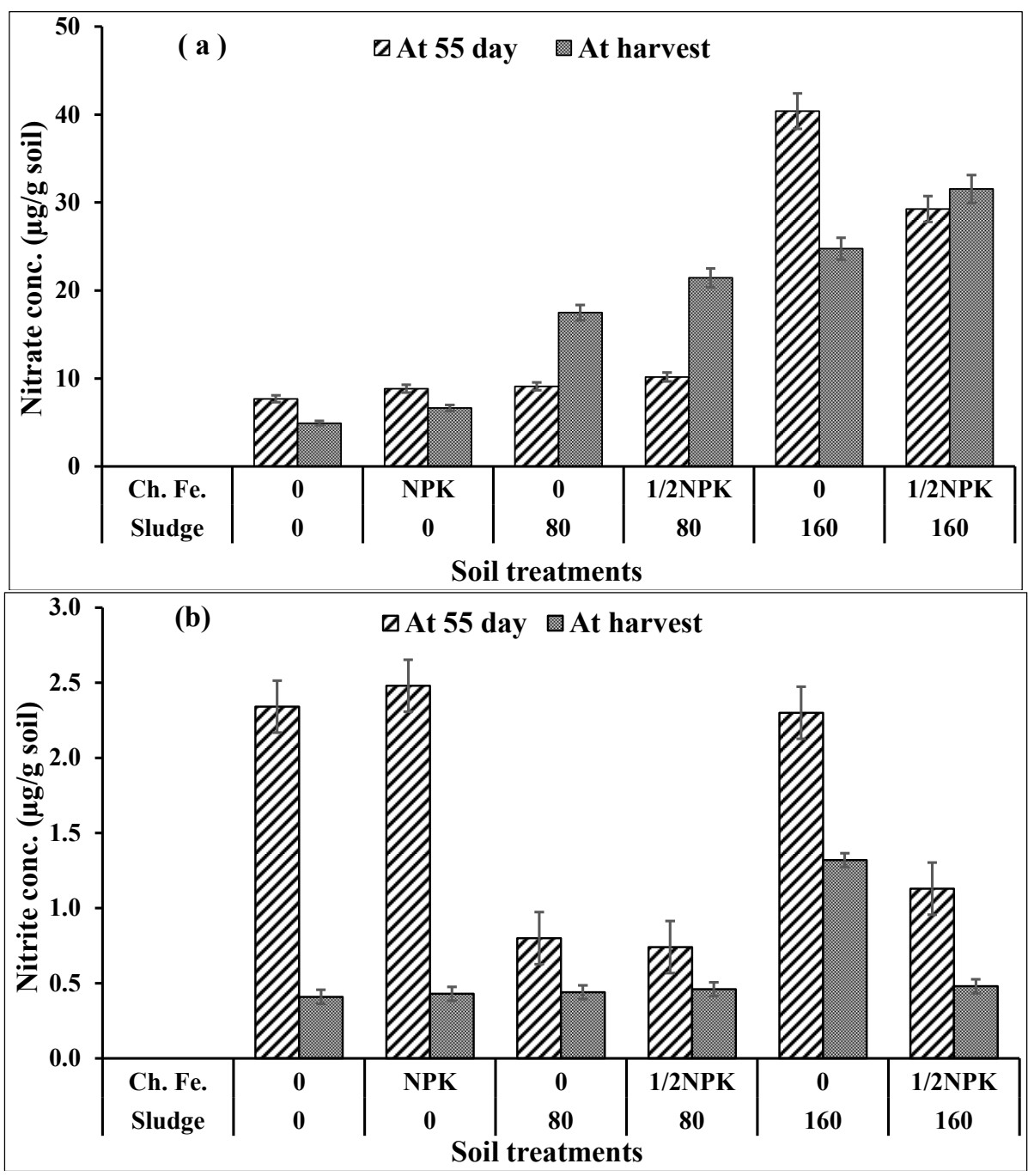

**Figure 2.** *Cont.*

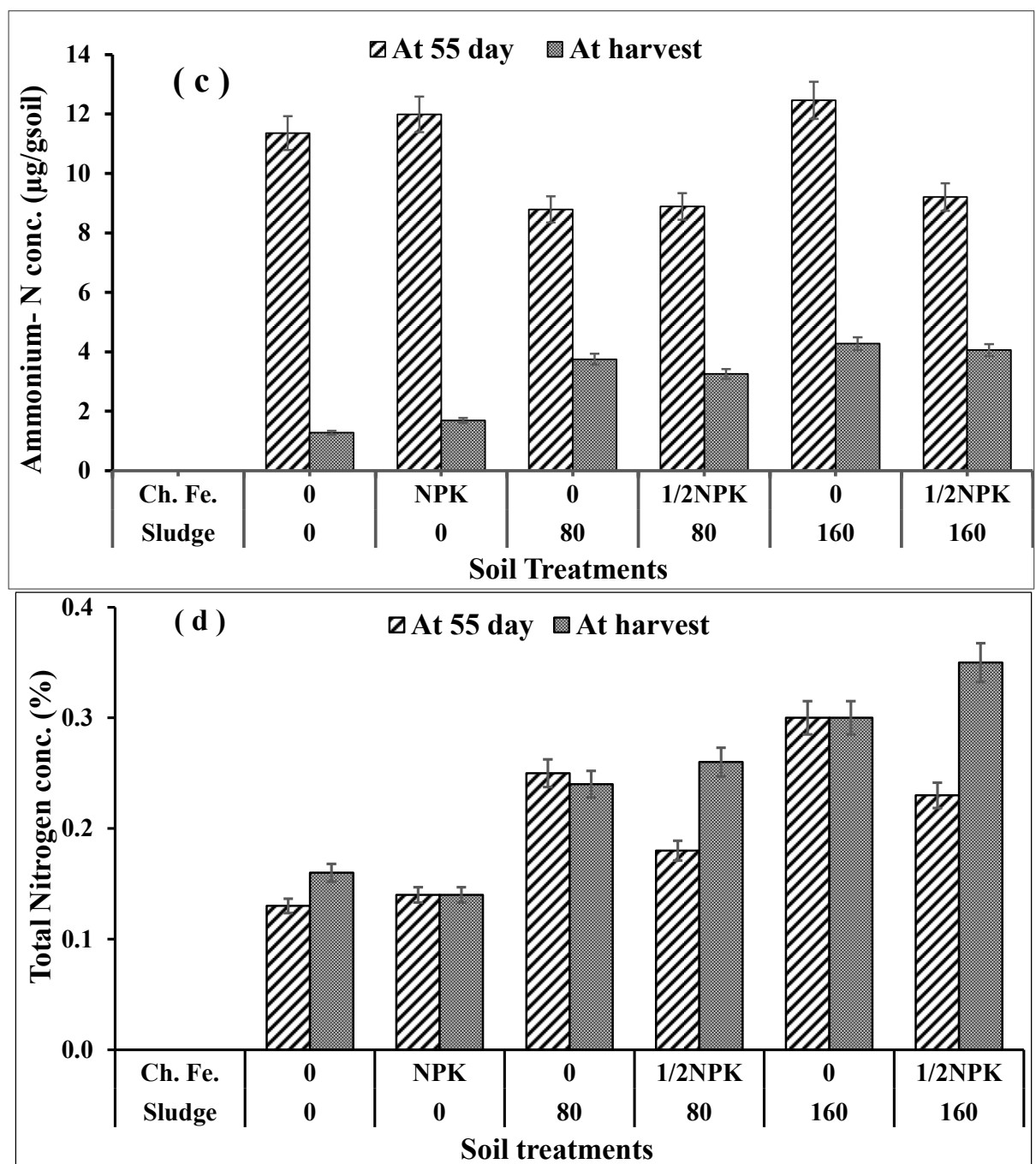

**Figure 2.** Effect of soil treatments on $NO_3^-$-N (**a**), $NO_2^-$-N (**b**), $NH_4^+$ -N (**c**), and total nitrogen (**d**) in faba bean soil after 55 days from seeding and after harvest.

After faba bean harvest, there were significant increases in the levels of $NO_3^-$ in sludge-amended soils relative to the control and NPK-CF soils. However, there were no significant effects of NPK-CF incorporated with sludge-amended soils on the levels of $NO_3^-$ in soils. Generally, the $NO_3^-$ levels of soils of the upper layer (0–15 cm) were significantly higher than those of the lower layer (15–30 cm). The data also showed that the concentration of $NO_2^-$ increased significantly only in sludge-amended soil at a sludge rate of 160 t/acre. The incorporated NPK-CF with sludge-amended soil significantly decreased the level of $NO_2^-$ in soil at a sludge rate of 160 t/acre. The distributions of $NO_2^-$ in soils were markedly variable and had no special trend. The levels of $NH_4^+$ increased significantly in the sludge-amended soils, relative to the control and NPK-CF

soils. Generally, the levels of $NH_4^+$ in soils decreased significantly with soil depth. The soil contents of total-N increased significantly in the sludge-amended soils relative to the control and NPK-CF soils. Generally, the levels of total-N in soils decreased significantly with soil depth, especially with sludge-amended soils.

Nitrogen mineralization of manures is not constant during the growing season and is affected by several factors; therefore, if maximum nitrogen mineralized is not synchronized with the maximum plant requirement with N uptake, a high amount of nitrogen mineralized through nitrification results in limited plant yield and growth [85]. Hence, synchronizing N mineralized from organic manures with crop N uptake and demand can maximize N uptake efficiency and minimize N loss during the crop growing season, meaning that time and rate of organic fertilizers are critical aspects in manure recommendations and management programs. In this case, Maguire and Alley [86] and Velasco et al. [87] suggested that split application of fertilizers is a good strategy to achieve high NUE and reduced N losses. It means that the high amount of manure should be applied exactly before the period of most rapid plant N uptake.

## 4. Conclusions

The data for corn shows marked variations in the response to NPK-CF with or without sludge application. Despite the occurrence of significant response to soil treatments as indicated by the yield of stover and grain, the corn crop varied markedly in response to soil treatments. There was a marked increase in the grain yield percentage relative to the biomass yield for corn. Moreover, the stover/grain ratio, which is an index of grain production, was approaching the unity for corn. In regard to faba beans, there were significant increases in the stover and grain yield with NPK-CF with or without residual effects of sludge. There was a higher response for the faba bean plant to residual effects of sludge, as indicated by grain production.

Soil OC levels were increased in soil after corn harvest at the higher sludge rate applied to the soil. The levels of OC in soils after crop harvest are proposed to be the residual carbon for the successive crops, faba beans after corn. The levels of SOC in soil samples collected after faba bean harvest was lower than those collected after 55 days from faba bean planting, especially in sludged soils. This points out the decreased residual effect of sludge in the soil for soil contents of OC.

The concentrations of $NO_3^-$, $NO_2^-$, $NH_4^+$, and total-N in soil samples collected after corn harvest are considered sources of the N residue from the harvested crop in sludge-amended soils. The same trend was found with faba bean soils as compared with those of corn soils. There is a marked increase in the levels of N forms in corn soils after crop harvest. There were increases in $NO_3^-$ and total-N and decreases in $NO_2^-$ and $NH_4^+$ levels in faba bean soils after crop harvest. Generally, there was marked depletion of N in soils after the harvest of plants grown on the residual sludge. Understanding the pattern and rate of N mineralized in organic-N-containing substances is essential to estimate application rate and managing nitrogen availability to improve N uptake efficiency and minimizing N loss via volatilization, leaching, and denitrification in crop production.

The analysis of the chemical composition of the sludge indicated that the heavy metal content did not exceed limit values, thus allowing for their application to the soil and use in agriculture or reclamation. On this basis, it is possible to understand the practical applicability of SS mixtures for agricultural use. In addition, the recovery of macro-elements from organic waste allows for the sustainable use of waste, and the macro-elements can be re-used in the production cycle. Thus, this research aligns with the European Union's Action Plan on the Circular Economy, which encourages the use of waste as fertilizers. Future work should also examine alternative doses and combinations of sludge and other organic fertilizer mixtures to establish optimal values in relation to the intended biomass utilization, support sustainable waste management, save chemical fertilizer application, and maintain environment and soil health.

**Author Contributions:** Conceptualization, H.E.A.E. and A.E.M.; Data curation, H.E.A.E., A.E.M., M.E.E.-S. and E.H.E.-G.; Formal analysis, H.E.A.E. and A.E.M.; Funding acquisition, H.E.A.E.; Investigation, H.E.A.E.; Methodology, H.E.A.E., M.E.E.-S., A.E.M. and E.H.E.-G.; Resources, H.E.A.E. and B.M.R.; Software, H.E.A.E. and M.E.E.-S.; Supervision, H.E.A.E.; Visualization, H.E.A.E.; Writing—original draft, H.E.A.E.; Writing—review & editing, H.E.A.E. and A.E.M. All authors have read and agreed to the published version of the manuscript.

**Funding:** This research was funded by Project number (TURSP-2020/139), Taif University, Taif, Saudi Arabia.

**Institutional Review Board Statement:** Not applicable.

**Informed Consent Statement:** Not applicable.

**Acknowledgments:** This work was carried out using the facilities and materials in Taif University Researches Supporting Project number (TURSP-2020/139), Taif University, Taif, Saudi Arabia.

**Conflicts of Interest:** The authors declare no conflict of interest.

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
