# Peer review of "Effect of Sewage Sludge Compost Usage on Corn and Faba Bean Growth, Carbon and Nitrogen Forms in Plants and Soil"

_agronomy, doi:10.3390/agronomy11040628_

Round 1
Reviewer 1 Report
Effect of Sewage Sludge Compost Usage on Corn and 2 Faba bean growth, Carbon and Nitrogen forms in 3 Plants and Soil
This paper examines the impact of sewage sludge and NPK applications on the growth of two crop varieties (faba and corn).
The results and discussion section is largely superficial, it describes the findings, but does not include sufficient detail of the mechanisms behind the observed impacts. There is not link to how the findings of this study can be used to inform or improve current practices.
In all this is an interesting study and suitable for publication subject to some editing.
- Although the manuscript is easy to understand, I would recommend a check of grammar.
- Check the formatting of nitrate and ammonium abbreviations (NO3- and NH4+).
Methods
- Formatting of Tables could be improved to better use space and presentation.
- Table 2 – the bottom line of the Table has been cut-off.
- There is an issue with the format for Section 2.6.
- Section 2.7 - which extractants and soil to extraction ratios were used for each analyte?
- It is not clear how many replicates were performed for each treatment or analysis.
Results & Discussion
- Figure captions need more detail, they should contain enough detail to be understood by the reader without having to refer back through the text.
- Figures do not include any error bars.
- Page 7, paragraph 1 – It is stated that N and P are within critical concentrations, but K is not, what are the critical concentrations?
- What does ‘ton/fed’ mean?
- There is a lack of detail with regard to the mechanisms or suspected mechanisms behind the observations, for instance
- The presentation of Figure 7 and Figure 8 should be improved.
Conclusion
- The conclusion could have been strengthened by including details of how these findings might be implemented, what is their significance with regard to agricultural practices.
- Add some information on how these findings contribute to or improve this field of knowledge.

Author Response
Effect of Sewage Sludge Compost Usage on Corn and Faba bean growth, Carbon and Nitrogen forms in Plants and Soil.
Respond to reviewer:
- Although the manuscript is easy to understand, I would recommend a check of grammar.
- Grammar had been checked and corrected
- Check the formatting of nitrate and ammonium abbreviations (NO3- and NH4+).
- Nitrate and ammonium abbreviations had been checked and corrected
Methods
- Formatting of Tables could be improved to better use space and presentation.
- Tables had been improved to better use space and presentation.
- Table 4 was exchanged in order with Table 5.
- Table 6 had been supplemented with corn leaves dry weight (g/one leaf) 65 days after seeding.
- Table 7 had been supplemented with faba bean leaves and stems dry weight (g/one leaf) 55 days after seeding.
- Table 9 had been added to show the effect of soil treatments on organic carbon in faba bean soil after 55 days from seeding and after crop harvest.
- Table 2 – the bottom line of the Table has been cut-off.
- Checked and corrected
- There is an issue with the format for Section 2.6.
- Format had been corrected
- Section 2.7 - which extractants and soil to extraction ratios were used for each analyte?
- Total and mineral nitrogen forms were extracted [extraction at 1:5 with 2 M potassium chloride (KCl)] (had been corrected in manuscript)
- It is not clear how many replicates were performed for each treatment or analysis.
- (The experiment was arranged in a randomized complete block design (RCBD) with four replicates.)
Results & Discussion
- Figure captions need more detail, they should contain enough detail to be understood by the reader without having to refer back through the text.
- Figures 1 and 2 had been deleted and data in tables 6 and 7, respectively are more enough to present results.
- Figures 3 and 4 had been deleted and data in table 8 are more enough to show the effect of soil treatments on organic carbon in corn soil 65 days after seeding and after crop harvest.
- Figures 5 and 6 had been deleted and replaced with table 9 to show the effect of soil treatments on organic carbon in faba bean soil after 55 days from seeding and after crop harvest.
- After previous deletion of figures (1-6), figures 7 and 8 had a new order and take the numbers 1 and 2.
- Figures do not include any error bars.
- Figures had been edited and supported with error bars
- Page 7, paragraph 1 – It is stated that N and P are within critical concentrations, but K is not, what are the critical concentrations?
- The paragraph had been edited (in the manuscript)
- What does ‘ton/fed’ mean?
- It means (t/acre), and it had been replaced with (t/acre)
- There is a lack of detail with regard to the mechanisms or suspected mechanisms behind the observations, for instance
- All results had been discussed again and supported with mechanisms or suspected mechanisms behind the observations, and compared with previous studies in publications.
- The presentation of Figure 7 and Figure 8 should be improved.
- Figure 7 and Figure 8 had a new order and take numbers 1 and 2, and had been improved.
Conclusion
- The conclusion could have been strengthened by including details of how these findings might be implemented, what is their significance with regard to agricultural practices.
- The conclusion had been edited and strengthened by including details of how these findings might be implemented, and their significance with regard to agricultural practices.
- Add some information on how these findings contribute to or improve this field of knowledge.
- Some information on how these findings contribute to or improve this field of knowledge had been added in the conclusion.
Reviewer 2 Report
attached file contains light editing via comments on the pdf

Author Response

(The authors gave the same response as above.)
